# Lateral Force and EMG Activity in Wide- and Narrow-Grip Bench Press in Various Conditions

**DOI:** 10.3390/sports11080154

**Published:** 2023-08-15

**Authors:** Michiya Tanimoto, Hiroshi Arakawa, Mauto Sato, Akinori Nagano

**Affiliations:** 1Graduate School of Health and Sports Science, Juntendo University, Inzai 299-5295, Chiba, Japan; 2Department of Physical Education, International Budo University, Katsuura 299-5295, Chiba, Japan; arakawa@budo-u.ac.jp; 3Graduate School of Sport and Health Science, Ritsumeikan University, Kusatsu 525-0058, Shiga, Japan; mauto1d0322@gmail.com; 4Faculty of Sport and Health Science, Graduate School of Sport and Health Science, Ritsumeikan University, Kusatsu 525-0058, Shiga, Japan; aknr-ngn@fc.ritsumei.ac.jp

**Keywords:** bench press, lateral force, electromyography, training experience, concentric and eccentric movement, muscle fatigue

## Abstract

The purpose of this study was to investigate the lateral force and contribution of shoulder horizontal adductor and elbow extensor muscles activity during wide- and narrow-grip bench press (BP) in various conditions, such as resistance-trained/non-trained, concentric/eccentric, and muscle fatigue/non-fatigue. We measured the lateral force on the bar and the electromyographic (EMG) muscle activity of pectoralis major (PM) and triceps brachii (TB) during 10 RM BP with wide grip (81 cm) and narrow grip (40 cm) in seven resistance-trained men and seven non-trained men. The all-reps average of the lateral-to-vertical force ratio both in resistance-trained and non-trained subjects was about 30% outward for the wide grip and about 10% inward for the narrow grip. The EMG contribution ratio PM/TB shows no significant differences between narrow and wide grip in all evaluated conditions except in non-trained subjects’ muscle fatigue eccentric phase. Both resistance-trained and non-trained subjects did not push the bar straight upward, and the EMG PM/TB was almost unchanged by hand width. The direction adjustment of the force on the bar that achieves almost the same muscle activity degree of the shoulder and elbow joints might be optimal BP kinetics.

## 1. Introduction

Many people performing bench press (BP) training may believe that they push the bar up with vertically upward force in BP, although in reality, they do not. In BP, the force exerted by each hand on the bar has an outward or inward lateral component [1,2,3]. Duffy et al. reported that the ratio of the lateral component to the vertical component averaged approximately 27% outward in the 1 RM and approximately 24% outward in the 80% 1 RM during BP movement [1]. In that study, the hand width condition for BP was applied at the participant’s discretion within the rules of competitive BP (less than 81 cm) [4]. Larsen et al. evaluated the direction of the BP bar pressing force under several hand width conditions and assessed the angle of the force vector pushing the bar at the sticking point of a concentric BP movement at 1 RM with a wide grip of 1.7 times the acromion width (average 71 cm), a medium grip of 1.4 times the acromion width (average 56 cm), and a narrow grip of approximately the acromion width (average 40 cm). The average force angle was 8.8° outward for the wide grip and 1.3° and 5.4° inward for the medium and narrow grips, respectively [2]. Mausehund et al. reported that the average lateral-to-vertical force ratio was 38% outward for the wide grip and 4% inward for the narrow grip in 80–85% 1 RM BP [3]. The lateral force on the bar in BP changes the force vector direction and might lead to the appropriate distribution of the moment arm length of the shoulder and elbow joints. The appropriate distribution of shoulder and elbow joint moments could increase the lifting (vertical) force in BP. The assumed mechanics are described later in the Section 4 using Figure 4.

Resistance training (RT) textbooks often state that the wider hand grip in BP increases the contribution of the pectoralis major (PM), and the narrower grip increases that of the triceps brachii (TB) [5,6]. However, several studies evaluating the electromyographic muscle activity (EMG activity) of individual muscles in BP do not support this as far as EMG is concerned [7,8]. There was no significant difference between the EMG activity of the PM and TB in wide-grip and narrow-grip BP under 6 RM conditions in competitive BP athletes [7]. The EMG activity of each muscle was not evaluated in the form of contribution ratio such as EMG of PM/EMG of TB; however, as far as the respective figures are concerned, it can be inferred that there was little difference in the ratios of EMG of PM and TB between the wide and narrow grips. These results indicate that the same level of activation of PM and TB EMG in the wide and narrow grips might lead to optimal kinetics in BP. It means that full exertion in both PM and TB might lead to maximum lifting force in BP. The ratios of EMG of PM and TB would depend on the bar-pushing direction in BP.

In this study, we compared the magnitude of the lateral force component and the contribution ratio of the EMG activity of the PM as muscle for shoulder horizontal adduction and the TB as muscle for elbow extension with wide and narrow grip 10 RM BP in various conditions. The lateral force component and EMG activity analysis were evaluated during concentric and eccentric movement phases, with the first 2 reps and the last 2 reps, in resistance-trained and non-trained subjects. The first 2 reps were taken as a non-fatigue and the last 2 reps as a maximal muscle fatigue phase. The magnitude of lateral force component in BP and the contribution ratio of PM and TB EMG activity in wide- and narrow-grip BP may vary depending on RT experience, concentric/eccentric phase, or degree of muscle fatigue.

BP movements in concentric and fatigue phases have a larger relative load to lifting force capacity than in eccentric and non-fatigue phases, so they might have a high need for optimal BP kinetics. Resistance-trained subjects might have a higher ability to efficiently lift heavy weights in BP than non-trained subjects. Many people might think that the direction of force pushing the bar in BP is vertically upward. So, we hypothesized that if the force direction adjustment leads to optimal kinetics in BP, the lateral force component might be larger, and differences in PM/TB EMG contribution between wide and narrow grips might be smaller in resistance-trained subjects in the concentric movement phase and in the last two reps than in non-trained subjects, and in the eccentric movement phase and in the first two reps.

## 2. Materials and Methods

### 2.1. Experimental Approach

Ten RM BP was performed by resistance-trained and non-trained subjects with a wide grip (81 cm hand width) and a narrow grip (40 cm hand width). The lateral force of the bar and EMG activity of the PM as horizontal shoulder adductor and the TB as elbow extensors were measured and evaluated per movement phase (concentric and eccentric movements) and per rep (first 2 reps and last 2 reps) with respect to the values obtained for the abovementioned grip types.

### 2.2. Subjects

The participants were composed of 14 healthy male adults, seven of whom were resistance-trained men with more than three years of RT experience and seven of whom were non-trained men. People under medical care for orthopedic or other diseases were excluded from this experiment. Subjects were recruited using posters on boards in the university. Resistance-trained subjects had 7.3 ± 5.3 years of RT experience (mean ± standard deviation). The age of the participants was 24.6 ± 6.5 yr for resistance-trained subjects and 23.4 ± 5.1 yr for non-trained subjects, their height was 171.2 ± 5.8 cm for resistance-trained subjects and 174.1 ± 6.5 cm for non-trained subjects, and their body mass was 69.8 ± 4.3 kg for resistance-trained subjects and 66.2 ± 10.3 kg for non-trained subjects.

### 2.3. BP Procedures

Ten RM BP were performed with a wide grip of 81 cm and a narrow grip of 40 cm hand width, and the lateral compression and tensile force of the bar and the EMG activity of PM and TB were evaluated for each movement phase and rep. The hand width upper limit is 81 cm in competitive BP rules [4]. The width of the acromion and the minimum measurement width of the prepared lateral compression and tensile force measurement bar is approximately equivalent to 40 cm. The lateral force component of the bar was measured using a strain gauge built into the bar. The cadence of movement was defined as one second up and two seconds down, and the range of movement was from full extension of the elbow joint to the point where the bar touched the chest. The cadence was guided using a metronome at a rhythm of 60 bpm. The subjects touched the barbell on the chest at a voluntary arbitrary position. Subjects were instructed to perform an open armpit form to avoid sagittal movement in narrow-grip BP. The two tests were performed on the same day with a 2 h break in between. The execution order of the two tests was random, and the trial order was equalized. The 10 RM trial of BP to determine the experiment load was performed more than a week before conducting the experiment. The trial load was set at a weight with which the participants were expected to be able to lift approximately 10 reps. Participants performed BP reps until failure. The experiment 10 RM load was calculated from the data of this trial loads and number of reps using the RM conversion table [9].

Subjects performed 10 RM BP in the laboratory with a temperature setting of 20 °C during afternoon–evening hours under the direction of the researcher familiar with resistance training science. Various data listed below were collected during 10 RM BP.

### 2.4. Lateral Tensile and Compressive Force Measuring Equipment

The bar for measuring the tensile and compressive forces in the longitudinal direction was constructed based on the shape of a powerlifting competition bar with a length and mass of 2200 mm and 20 kg, respectively (Takei Scientific Instruments Co., Ltd., Niigata, Japan) (Figure 1). A steel sleeve was mounted over the bar grip and connected to a tensile/compression gauge installed inside the plate mounting area. Linear bushings were used inside the sleeve at the contact points to minimize the effects of friction. An amplifier (TD-700T, TEAC, Tokyo, Japan) amplified the voltage data from the tensile/compression gauge, which was then fed into a data acquisition system (Power Lab/16SP, AD Instruments, Bella Vista, NSW, Australia) and smoothed using a low-pass filter at 20 Hz by analysis software (Lab chart, AD Instruments, Australia). The lateral tensile and compressive forces of the bar were calibrated prior to the measurements. Calibration was performed to identify the measured values from the linear relationship between voltage and load using 0 kg, 25 kg, and 50 kg weights. Errors in the repeatability tests were within 3% under the 80 kg loaded condition and within 0.5% under the unloaded condition.

### 2.5. Lateral Force Evaluation

The lateral tensile and compressive force acting on the bar was evaluated using the lateral tensile and compressive force/vertical force. The lateral force outward and inward directions were defined as positive (+) and negative (−), respectively. The vertical force was calculated using the value of a 3-axis accelerometer (DL-111, S&ME, Tokyo, Japan) attached at the end of the barbell shaft with adhesive tape and assuming that the vertical load was equally applied to the left and right hands. The Z-axis of a 3-axis accelerometer on the bar was adjusted in the vertical direction by visual observation. The vertical force calculation formula is as Equation (1) below. The acceleration data were recorded in a data acquisition system (Power Lab/16SP, AD Instruments, Australia) and smoothed with a low-pass filter at 20 Hz using analysis software (Lab chart, AD Instruments, Australia).
Vertical force(N) = BP (kg)/2 × (9.81 ± vertical acceleration (m/s/s))(1)

### 2.6. Electromyography (EMG)

Electromyograph (EMG) signals were recorded from the dominant pectoralis major sternocostal part (PM) and triceps brachii lateral head (TB). PM and TB are primary muscles in BP [10]. The subjects’ skin was washed with alcohol before placing the electrodes. Bipolar surface electrodes (Vitrode F, Nihon Kohden, Tokyo, Japan) were placed over the belly of the muscles with a constant interelectrode distance of 20 mm. The procedures and placement were performed in accordance with the recommendations of SENIAM [11]. The PM is not listed on SENIAM, and so for this muscle, the electrode was placed horizontally from the mid-sternum position. The EMG signals were amplified, fed into a full-wave rectifier through both low (30 Hz) and high (1 kHz) cut filters using analysis software (Lab chart, AD Instruments, Australia), and stored by using a data acquisition system (Power Lab/16SP, AD Instruments). The mean rectified EMG signals from each muscle were normalized using EMG recordings from the subjects’ three seconds of maximal isometric voluntary contraction (MVIC) of each muscle. Shoulder horizontal adduction (90-degree shoulder abduction and 0-degree horizontal adduction angle: PM) and elbow extension (90-degree elbow flexion angle and 90-degree shoulder flexion angle: TB) were used. The ratio of EMG between PM and TB (EMG PM/TB) was calculated from the quotient of the mean rectified EMG normalized by MVIC of PM and that of TB.

### 2.7. Identification of the Concentric and Eccentric Movement Phase

A flexible goniometer (SG110, Biometrics, Newport, UK) was affixed to the proximal and distal elbow joints 10 cm from the elbow joint, respectively. The data from the flexible goniometer were acquired using a data acquisition system (Power Lab/16SP, AD Instruments, Australia) and smoothed with a low-pass filter at 20 Hz using analysis software (Lab chart, AD Instruments, Australia). The elbow joint angular velocity was calculated using the time difference of the smoothed angular data. The maximum and minimum values of the elbow joint angle data for each trial (that is, the time when the elbow joint angular velocity was zero) were determined as the time boundary between the concentric and eccentric movement phases.

### 2.8. Statistical Analyses

The results are presented as mean ± SD. A paired t-test was used to assess differences in the following three types of comparisons: in 10 RM BP load between two grip widths (wide and narrow); in lateral-to-vertical force ratio between two grip widths, two movement phases (concentric and eccentric) and two repetition phases (first 2 reps and last 2 reps); and in EMG activity in PM, TB and PM/TB between two grip widths. A two-way analysis of variance (test muscle × repetition phase) was performed to compare changes in first 2 and last 2 reps of EMG activity in PM and TB, respectively, and determine the interaction. When an interaction was found, a paired t-test was performed to compare changes in first 2 and last 2 reps of EMG activity in same muscle. We performed a parametric analysis assuming that the data for each measurement item were normally distributed. All calculations were performed using SPSS (version 27.0; SPSS Inc., Chicago, CA, USA). The statistical significance level was set at *p* < 0.05. The results are presented as mean ± SD. Effect size (ES) of each comparison was determined as Cohen’s d. An ES of <0.2 was considered trivial, 0.2–0.5 was considered small, 0.5–0.8 was considered medium, and >0.8 was considered large [12]. Assuming an effect size delta of 0.9, the statistical power of the paired t-test with 7 subjects is calculated to be 0.52.

## 3. Results

### 3.1. Estimated 10 RM Loads and the Number of Repetitions at These Loads

The load of 10 RM in resistance-trained subjects was significantly larger for the wide grip than for the narrow grip (wide: 81.3 ± 11.0 kg vs. narrow: 65.9 ± 10.7 kg, *p* < 0.01, d = 1.39). In non-trained subjects, this was significantly larger for the wide grip than for the narrow grip (wide: 39.2 ± 5.2 kg vs. narrow: 37.5 ± 5.2 kg, *p* < 0.05, d = 0.33). The number of repetitions until failure to lift in resistance-trained subjects 10 RM BP was 10.1 ± 1.1 for the wide grip and 10.1 ± 0.7 for the narrow grip. In non-trained subjects, this was 10.1 ± 1.7 for the wide grip and 10.4 ± 1.9 for the narrow grip. There was no significant difference in the number of repetitions performed by each hand width for either resistance-trained or non-trained subjects.

### 3.2. The Ratio of Lateral-to-Vertical Force

The lateral force exerted on the bar during BP was evaluated relative to the vertical force exerted on the bar. The all-reps lateral force average for resistance-trained subjects was +28.8 ± 16.3% and +27.0 ± 11.6% in the concentric and eccentric phases for the wide grip, respectively, and −17.3 ± 9.5% and −10.7 ± 10.7% in the concentric and eccentric phases for the narrow grip (+: outward, −: inward). For non-trained subjects, this was +38.4 ± 9.6% and +34.2 ± 6.9% in the concentric and eccentric phases for the wide grip, and −14.1 ± 11.0% and −12.1 ± 6.2% in the concentric and eccentric phase for the narrow grip (Figure 2). There were significant differences between wide and narrow grips in all conditions. The individual values in resistance-trained and non-trained subjects were plus (outward) for the wide grip and minus (inward) for the narrow grip in all participants.

Comparing the concentric and eccentric phases, the absolute values of all-reps lateral force average in the narrow grip of resistance-trained subjects and the wide grip of non-trained subjects were significantly smaller in the eccentric phase than in the concentric phase (resistance-trained subjects’ narrow grip: *p* < 0.01, d = 0.69; non-trained subjects’ wide grip: *p* < 0.01, d = 0.55). Although there was no statistically significant difference in the wide grip of resistance-trained subjects and in the narrow grip of non-trained subjects, the absolute values were slightly smaller in the eccentric phase than in the concentric phase. In other words, overall, the force vector was closer to vertically upward in the eccentric phase (Figure 2).

In the comparison between non-fatigue and fatigue phases, the absolute value of lateral force in the last 2 reps was significantly increased compared to the first 2 reps in the wide-grip concentric movement phase of non-trained subjects (*p* < 0.05, d = 2.22), and there was a slight increase in the overall trend, except in the resistance-trained subjects’ wide-grip eccentric phase. The absolute values of both the first 2 and last 2 reps of resistance-trained and non-trained subjects were smaller in the eccentric phase than in the concentric phase, and there were significant differences in the resistance-trained subjects’ first 2 narrow-grip reps (*p* < 0.01, d = 0.82), last 2 wide-grip reps (*p* < 0.05, d = 0.28), and non-trained subjects’ last 2 wide-grip reps (*p* < 0.05, d = 0.55) (Figure 3).

### 3.3. EMG Activity

The EMG activity (mean rectified EMG/MVIC) and EMG PM/TB at the all-reps averages, and the first 2 and last 2 reps are shown in Table 1 and Table 2. Although they were not statistically significant, the EMG PM/TB at the all-reps averages for both resistance-trained and non-trained subjects were slightly smaller for the narrow grip than for the wide grip during the concentric phase. Both resistance-trained and non-trained subjects’ values tended to be smaller in the narrow grip than in the wide grip during the eccentric phase (RTP: *p* = 0.11, d = 0.32; NTP: *p* = 0.07, d = 1.05). Focusing on non-fatigue and fatigue conditions, EMG PM/TB in the last 2 reps during the eccentric movement phase tended to be lower in the narrow grip than in the wide grip for resistance-trained subjects (*p* = 0.12, d = 0.49) and significantly lower in non-trained subjects (*p* < 0.05, d = 0.93).

When comparing the first 2 and last 2 reps, an interaction effect was observed in the changes in EMG activity of PM and TB under any conditions. The EMG activity of PM was significantly higher in the last 2 reps than in the first 2 reps under all conditions (approximately +20–40%). In contrast, the EMG activity of TB exhibited small differences between the first 2 and last 2 reps.

## 4. Discussion

The all-reps average of lateral-to-vertical force ratio in the concentric phase was 27.6 ± 13.1% outward with the wide grip and 12.9 ± 10.8% inward with the narrow grip in resistance-trained subjects. As for non-trained subjects, those were 35.5 ± 7.5% outward with the wide grip and 13.1 ± 7.5% inward with the narrow grip. These results approximately agreed with the results of previous studies [1,2,3]. It may seem inefficient to have a lateral force component to the vertical gravity load in BP, i.e., the oblique force at an angle θ to the vertical appears to waste force by a factor of (1 − cos θ). However, the magnitude of the vertical force exerted in BP could be increased by adjusting the direction of the force pushing the bar.

We examined a BP kinetics calculation that uses a simplified 2D static model (Figure 4), which assumes that the mass of the upper and forearm segments and the acceleration of the bar and each segment are zero, the upper and forearm segments are the same length, and the movement is only in the horizontal plane. In the case of a posture with a 0° horizontal abduction at the shoulder joint and 90° at the elbow joint, when the force is applied to the bar in the vertically upward direction, only the shoulder horizontal abduction moment is exerted, and the elbow extension moment is zero. On the other hand, when the force to the bar has a lateral outward component, the moment arm is divided between the shoulder and the elbow joints. The calculation is as follows: Although limited to the case of 0° horizontal abduction of the shoulder joint (upper arm horizontal to the ground), vertical force on the bar is proportional to the sum of the shoulder horizontal adduction and elbow extension moments; that is, the force vector direction in which the moment distribution at the shoulder and elbow joints matches the respective maximal muscle force ratios maximizes this calculated lift force, regardless of the hand width. If the direction of force in BP was subconsciously optimized to maximize the vertical force by adjusting the distribution of the shoulder and elbow joint moment according to their strength ratio, the contributions of shoulder horizontal adduction and elbow extension muscle activity would have little difference related to hand width. Larsen et al. reported that the moment arm length of the elbow joint and that of the shoulder joint was almost the same among wide-, medium-, and narrow-grip BP at sticking point [2]. If the direction of force in BP approaches vertically upward, the contributions of shoulder horizontal adduction muscle activity would become larger in the wide grip, and the contributions of elbow extension muscle activity would become larger in the narrow grip.

Assessment by EMG PM/TB of the contribution of PM as a horizontal shoulder adductor and TB as an elbow extensor showed a little difference associated with hand width in almost all experimental conditions. No significant difference in EMG PM/TB between the wide and the narrow grip was observed for the all-reps average both in resistance-trained and non-trained subjects (Table 1 and Table 2). It would be surmised that the BP movements were close to optimal for increasing lifting force both in resistance-trained and non-trained subjects. Several previous studies examined the relationship between the EMG activity of the PM, TB, and other muscles depending on the BP hand width [7,8]. The EMG activity of each muscle was not evaluated in the form of contribution ratio as EMG of PM/EMG of TB in previous studies; however, as far as the respective figures are concerned, it can be inferred that there was a little difference in the ratios of EMG of PM and TB between the wide and narrow grips. Using the assessment by EMG PM/TB constitutes the significance of this study.

Although the differences were slight, the mean values of the EMG PM/TB for the wide grip were greater than those for the narrow grip under almost all comparative conditions. The contribution of the PM activity was slightly higher with the wide grip, and that of the TB was slightly higher with the narrow grip. This trend was stronger in non-trained subjects. It is possible that the RT experience further optimized the movement to increase the lifting force and reduced the difference between the contribution of PM and TB with hand width. Based on the premise that many people are conscious of pushing the bar vertically in BP, non-trained subjects with low BP lifting skill would be expected to have a smaller lateral force component in BP, but this was not the case in the wide grip. The lateral force component in wide-grip BP was high in non-trained subjects. The magnitude of the lateral force component might also be related to the difference in the muscle strength and the ability to exert muscle strength in BP movement of shoulder horizontal adduction and elbow extension. The EMG activity/MVIC of PM muscle appeared to be low for non-trained subjects. This might be related to a high lateral force component in wide-grip BP in non-trained subjects. Based on mechanical theory, a smaller shoulder joint horizontal adduction moment compared to the shoulder extension moment results in a higher lateral force component in wide-grip BP.

In addition to the RT experience, the phase of the movement (concentric/eccentric) and degree of muscle fatigue (first 2 reps/last 2 reps) may also affect the ratio between the magnitude of the lateral force component and EMG activity. Strength in concentric movements is lower than that in eccentric movements. One BP study reported that young men with RT experience had approximately 40% greater force in eccentric than in concentric movements [13]. Concentric movements may be more prone to movement optimization than eccentric movements because of their higher relative load intensity. Concentric movement in BP was predicted to have a larger lateral force component, leading to moment sharing between the shoulder and elbow joints, and have smaller EMG PM/TB differences in hand width. The present study observed the approximate predicted changes for both resistance-trained and non-trained subjects, although the differences were not large. The absolute value of the lateral force component was significantly higher in the concentric phase than in the eccentric phase for the narrow grip of resistance-trained subjects and the wide grip of non-trained subjects in the all-reps average. It means the pushing direction was closer to vertically upward in the eccentric phase. The EMG activity of PM/TB tended to be lower in the narrow grip than in the wide grip for both resistance-trained and non-trained subjects during the eccentric phase for all reps and was significantly lower in the narrow grip than in the wide grip during the eccentric phase for the last 2 reps for non-trained subjects.

Similar to the concentric movement phase, in the last 2 reps, when muscle fatigue progresses, and relative load intensity increases, it would be predicted that the lateral force component will increase and the difference in EMG PM/TB by hand width will decrease as compared with in the first 2 reps. As for the lateral force component, it showed the predicted changes in the concentric movement phase. The absolute value of the lateral force component during the concentric phase for the wide grip was significantly higher in the last 2 reps than in the first 2 reps for non-trained subjects. On the other hand, the differences in EMG PM/TB by hand width were almost the same between the last 2 and the first 2 reps for both resistance-trained and non-trained subjects in the concentric phase, and the differences with hand width appeared to widen more in the last 2 reps than in the first 2 reps in the eccentric phase. These results were not the predicted changes. However, it is difficult to interpret these results because EMG activity is affected by the accumulation of muscle fatigue up to that point. This is a task to be addressed in the future.

The lateral force applied to the bar during BP is not small and reached about 40% of the vertical force in this study’s condition. Thus, the amount of friction on the bar is considered one of the important factors that affects the maximum lifting load and ease of performing BP for both resistance training beginners and seniors. Some bars are easy to lift, while others are difficult to lift, which may be related to the amount of friction on the bar. This may explain why non-slip powder is applied during BP in competitions. To increase the effectiveness of BP training, it may be effective to increase friction on the bar.

In the present study, 10 RM load was significantly lower in the narrow grip (40 cm hand width) than in the wide grip (81 cm hand width) both in resistance-trained and non-trained subjects. A previous study also reported that the narrower the hand width, the lower the load lifted [14]. In competition BP, the 81 cm hand width limit is often employed by athletes to shorten the vertical lifting distance [4]. In terms of the amount of using load, it can be said that a smaller hand width reduces the burden on joints and other parts of the body. Although there are various risk factors for BP injury other than those associated with the using load magnitude [15], a narrow grip may be an effective choice for reducing the risk of injury.

In addition, a smaller hand width increases the range of vertical motion, which increases the range of motion (ROM) of the shoulder and elbow joints. ROM size is one of the important factors affecting the hypertrophic effect of RT. Some studies using full-range RT and partial-range RT at the same RM have shown that the former had a greater effect on muscle hypertrophy than the latter [16,17]. Full-range RT in elbow flexion exercise resulted in a smaller load for the same RM, although the total mechanical work increased [18]. Mechanical work volume in RT is indicated as one of the important elements that leads to hypertrophy [19]. In addition, as the ROM increases, the scope of operation includes areas of greater muscle elongation. Muscle damage due to eccentric contraction is greater with increased muscle elongation [18,20]. Muscle damage from eccentric contractions is thought to be one of the effective stimuli that induces muscle hypertrophy [21]. If there is a little difference in the contribution of the working muscles of the shoulder and elbow joints depending on the hand width, performing a BP with a hand width that allows for a large ROM with a low load capacity may be an effective choice for a training program because of its safety and hypertrophic effect.

Although not the main focus of this study, there were two results in which specific features were observed. Regardless of hand width and RT experience, EMG activity in the PM increased significantly in the last 2 reps compared to the first 2 reps, whereas no significant change was observed in the TB (Table 1 and Table 2). EMG activity increases with the progression of reps due to muscle fatigue in single-joint exercise RT [22]. In BP, the changes in EMG activity suggest that the stimulus for muscle fatigue is greater for PM than for TB, regardless of hand width. BP exercise alone may be insufficient to provide muscle fatigue stimulation to TB. In addition, the EMG activity/MVIC of PM muscle appeared to be low for non-trained subjects. Non-trained subjects might not be able to exert sufficient shoulder horizontal adduction moment during BP. It might be that BP training experience makes PM become fully active during BP. Also, this might be partly related to the results of the lateral force component for non-trained subjects. Based on mechanical theory, the greater the moment exerted at the elbow joint relative to that at the shoulder joint, the higher the lateral component of the force vector in the wide grip and the lower the lateral component in the narrow grip.

This study has some experimental limitations. Under the narrow-grip condition in this study, the diameter of the grip part was relatively large (50 mm) compared to that of normal bars (28 mm) because of the measuring device contained inside. It is possible that the difference in the diameter of the grip may have affected the results. EMG activity in the last two reps is affected by the accumulation of muscle fatigue up to that point. EMG signals are affected by the degree of muscle shortening and subcutaneous movement of the muscle and make it harder to follow the values. These are limitations of this study.

## 5. Conclusions

The force to push the bar during BP has a lateral component, and the difference in the contribution of the shoulder horizontal adductor and elbow extensor muscles depending on the hand width is small both in resistance-trained and non-trained subjects. When selecting the hand width in BP, rather than focusing on the target training muscle, it would be effective to focus on other objectives such as safety, muscle hypertrophy/strengthening, and movement specificity. Also, it would be important to increase the frictional force of the bar to exert a sufficient lateral force component. The findings of this study may not only apply to BP but also to other RT events, such as overhead press and lat pulldown, which may have “lateral internal forces on the bar”.

## Figures and Tables

**Figure 1 sports-11-00154-f001:**
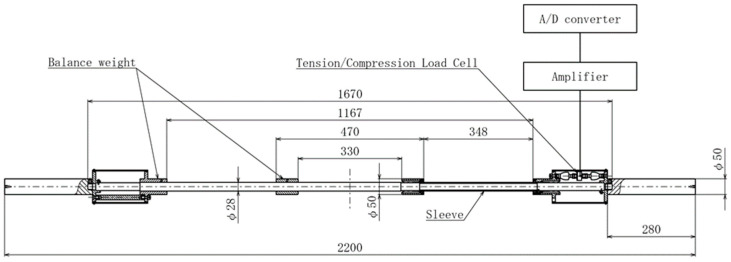
Device to measure lateral tension and the compression force of the bar. The tension and compression load cell is inside the bar. Linear bushings are used inside the sleeve at the contact points to minimize the effects of friction.

**Figure 2 sports-11-00154-f002:**
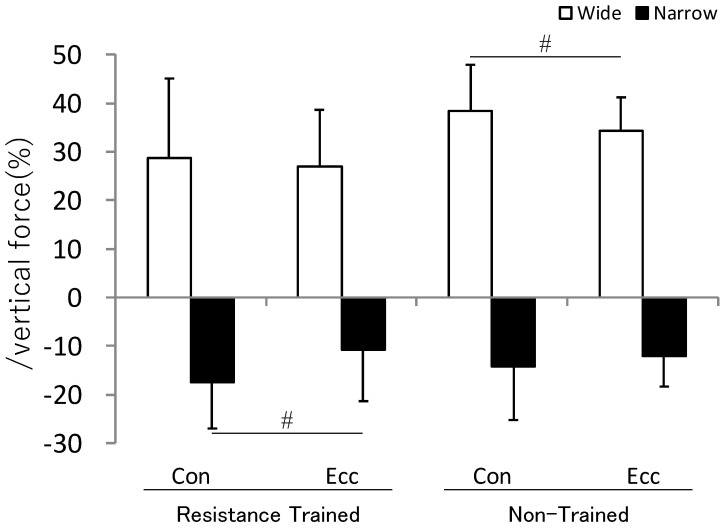
Lateral force acting on the bar at the all-reps average. Lateral force was evaluated as %vertical force. Values are mean ± SD. #: Statistical difference between the concentric and eccentric movement (*p* < 0.05). Although significant difference marks in this figure are omitted, there are significant differences between wide and narrow grips in all conditions. Con: concentric movement phase, Ecc: eccentric movement phase.

**Figure 3 sports-11-00154-f003:**
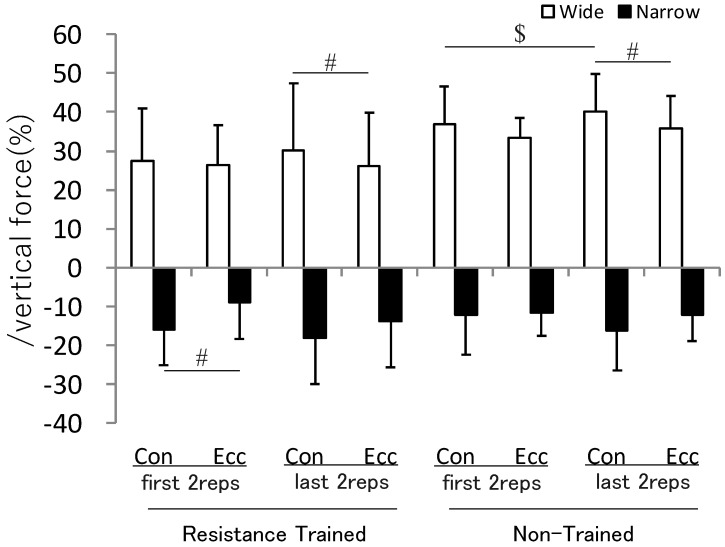
The lateral force acting on the bar during the first and last 2 reps. Values are mean ± SD. #: Statistical difference between the concentric and eccentric movement (*p* < 0.05). $: Statistical difference between first 2 reps and last 2 reps (*p* < 0.05). Although significant difference marks in this figure are omitted, there are significant differences between the wide and narrow grips in all conditions. Con: concentric movement phase, Ecc: eccentric movement phase.

**Figure 4 sports-11-00154-f004:**
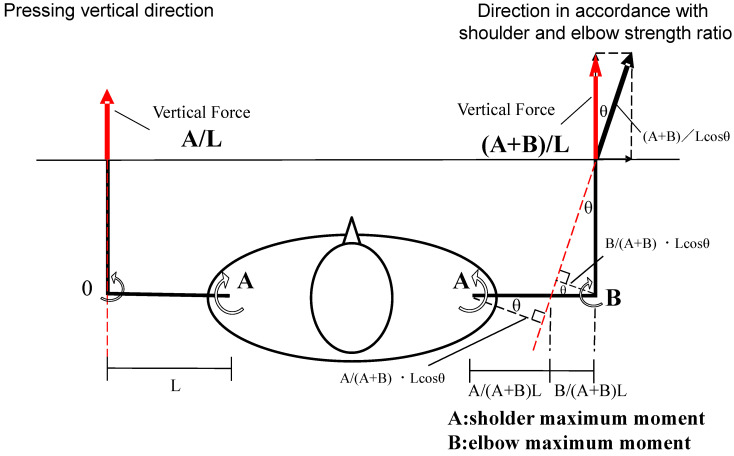
Joint moment calculation with simplified 2D static model. In the above posture, only the shoulder joint exerts moment, and the elbow moment is zero when pushed vertically upward (the left side illustration). When pushing in the direction that adjusts the distribution of the shoulder and elbow joints according to their muscle strength ratio (the right-side illustration), raising force is calculated to be (A + B)/A times the left-side vector illustration.

**Table 1 sports-11-00154-t001:** EMG activity of PM, TB/MVIC, and PM/TB in 10 RM BP in resistance-trained subjects.

Resistance- Trained	PM				TB				PM/TB		Cohen’s d (Wide vs. Narrow)		
	Wide		Narrow		Wide		Narrow		Wide	Narrow	PM	TB	PM/TB
all reps													
whole	0.67 ± 0.32		0.62 ± 0.26		0.70 ± 0.34		0.67 ± 0.23		1.11 ± 0.55	1.01 ± 0.45	0.20	0.11	0.20
concentric	0.98 ± 0.41		0.94 ± 0.37		0.93 ± 0.42		0.90 ± 0.29		1.19 ± 0.54	1.14 ± 0.48	0.10	0.10	0.10
eccentric	0.48 ± 0.26		0.43 ± 0.19		0.55 ± 0.31		0.54 ± 0.22		1.07 ± 0.65	0.89 ± 0.44	0.26	0.05	0.32
first 2 reps													
whole	0.56 ± 0.26		0.53 ± 0.22		0.69 ± 0.35		0.65 ± 0.23		0.94 ± 0.46	0.87 ± 0.33	0.13	0.15	0.19
concentric	0.84 ± 0.37		0.81 ± 0.32		0.86 ± 0.40		0.84 ± 0.27		1.14 ± 0.55	1.02 ± 0.40	0.11	0.06	0.24
eccentric	0.41 ± 0.22		0.39 ± 0.17		0.59 ± 0.32		0.55 ± 0.23		0.79 ± 0.42	0.76 ± 0.30	0.10	0.16	0.09
last 2 reps													
whole	0.77 ± 0.35	$	0.70 ± 0.28	$	0.69 ± 0.31		0.69 ± 0.23	$	1.24 ± 0.62	1.11 ± 0.51	0.25	0.01	0.27
concentric	1.02 ± 0.42	$	1.00 ± 0.41	$	0.92 ± 0.39	$	0.90 ± 0.26	$	1.22 ± 0.53	1.20 ± 0.53	0.05	0.06	0.07
eccentric	0.56 ± 0.32	$	0.47 ± 0.22	$	0.48 ± 0.20	$	0.53 ± 0.23		1.38 ± 0.97	1.02 ± 0.54	0.36	0.22	0.49

Values are mean ± SD; n = 14 (7 for each group). $: Significant difference between last 2 reps and first 2 reps (*p* < 0.05).

**Table 2 sports-11-00154-t002:** EMG activity of PM, TB/MVIC, and PM/TB in 10 RM BP in non-trained subjects.

Non- Trained	PM					TB		PM/TB			Cohen’s d (Wide vs. Narrow)		
	Wide			Narrow		Wide	Narrow	Wide		Narrow	PM	TB	PM/TB
all reps													
whole	0.47 ± 0.16			0.41 ± 0.18		0.81 ± 0.20	0.81 ± 0.22	0.57 ± 0.07		0.50 ± 0.11	0.39	0.14	0.74
concentric	0.65 ± 0.32			0.60 ± 0.35		1.15 ± 0.28	1.13 ± 0.33	0.55 ± 0.15		0.53 ± 0.21	0.17	0.07	0.14
eccentric	0.36 ± 0.12	*		0.29 ± 0.12		0.63 ± 0.15	0.64 ± 0.16	0.57 ± 0.07		0.46 ± 0.13	0.62	0.08	1.05
first 2 reps													
whole	0.38 ± 0.12			0.35 ± 0.15		0.74 ± 0.16	0.76 ± 0.22	0.50 ± 0.06		0.45 ± 0.10	0.23	0.11	0.55
concentric	0.51 ± 0.27			0.49 ± 0.29		1.00 ± 0.23	1.02 ± 0.33	0.49 ± 0.15		0.47 ± 0.19	0.08	0.06	0.14
eccentric	0.29 ± 0.11			0.27 ± 0.11		0.62 ± 0.11	0.64 ± 0.18	0.47 ± 0.11		0.43 ± 0.15	0.22	0.20	0.28
last 2 reps													
whole	0.54 ± 0.17	*	$	0.47 ± 0.22	$	0.83 ± 0.16	0.80 ± 0.17	0.65 ± 0.10		0.58 ± 0.15	0.37	0.19	0.53
concentric	0.72 ± 0.29		$	0.69 ± 0.34	$	1.17 ± 0.22	1.16 ± 0.28	0.62 ± 0.18		0.59 ± 0.20	0.12	0.07	0.15
eccentric	0.41 ± 0.13	*	$	0.32 ± 0.14	$	0.59 ± 0.12	0.58 ± 0.11	0.68 ± 0.09	*	0.55 ± 0.18	0.66	0.10	0.93

Values are mean ± SD; n = 14 (7 for each group). *: Significant difference between wide and narrow grips (*p* < 0.05). $: Significant difference between last 2 reps and first 2 reps (*p* < 0.05).

## Data Availability

Not available.

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
