# Peer review of "Lateral Force and EMG Activity in Wide- and Narrow-Grip Bench Press in Various Conditions"

_sports, 2023, doi:10.3390/sports11080154_

Round 1

Reviewer 1 Report

Congratulations to the authors. But some minor adjustments need to be made;

Please explain the research design of the study.

-The sample size calculation should be added to the method section.

-The data collection process cannot be detailed in method section. Where, by whom, and in what time of day the data were collected. It should be detailed.

-Inclusion and exclusion criteria from the study should metioned in the methods section.

-Which sampling method was used in the selection of the sample group? Please explain sample selection

-Which test was used to determine the normality of the data?

-Please write and develop practical applications section of the study.

-Please write limitations of the study.

Some minor revision should be made by the authors in all section by the author

Author Response

Thank you for reviewing our manuscript.

Kind regards.

Reviewer 2 Report

Overall not bad work, but needs a variety of improvements and modifications

line 20 report the mean +-SD values here.

Clarify throughout what you mean by horizontal, due to the nature of the bench press you need to move the bar horizontally when lowering it down to your chest and your definition of the axis of interest can be hard t follow. 

line 48-49 clarify this sentence more so

line 67-69 clarify this sentence as well. 

line 71 "concentric/eccentric phase"

Methods in general, what was the subjects wingspan and acromion width since this will influence arm and shoulder angles. where on the chest did they touch the barbell to?

line 111 "what does trial order was even" mean?

also "the 10RM BP"

Don't be afraid to insert the table of how you came to the calculations for 1RM prediction and more on the warm up and testing max values. 

Report the bar length in cm or m

line 119 remove "of the bar"

What type of filter did you use for your signal? butterworth? etc.?

Did you abrade the skin for EMG placement? give better anatomical landmark information about where the electrodes were placed

line 159 change to "angle"

line 190 and 192 change how you start these sentences, reads awkwardly.

line 223 and 231 label your graphs more clearly right now they read poorly

when discussing the emg activity be sure to discuss how as muscle bellys move the signal will change and makes it harder to follow the values. 

table 1 and 2 format your footnote to fit in the table

line 263 remove "in"

line 266-268 clarify this sentence more so, what do you mean here? how is this force being "wasted"?

Line 269-289 this paragraph needs references to back up the claims and points brought up.  The same thought applies to line 359-365.

Line 412 why is this important? Explain the logic more thoroughly here. 

Overall good writing, just a few grammatical errors and spelling mistakes. 
